# Learning Graph-embedded Key-event Back-tracing for Object Tracking in Event Clouds

**Zhiyu ZHU, Junhui HOU**,* **Xianqiang LYU**
Department of Computer Science, City University of Hong Kong
zhiyuzhu2-c@my.cityu.edu.hk, jh.hou@cityu.edu.hk,
xianqialv2-c@my.cityu.edu.hk

## Abstract

Event data-based object tracking is attracting attention increasingly. Unfortunately, the unusual data structure caused by the unique sensing mechanism poses great challenges in designing downstream algorithms. To tackle such challenges, existing methods usually re-organize raw event data (or event clouds) with the event frame/image representation to adapt to mature RGB data-based tracking paradigms, which compromises the high temporal resolution and sparse characteristics. By contrast, we advocate developing new designs/techniques tailored to the special data structure to realize object tracking. To this end, we make the *first* attempt to construct a new end-to-end learning-based paradigm that *directly* consumes event clouds. Specifically, to process a *non-uniformly* distributed large-scale event cloud efficiently, we propose a simple yet effective density-insensitive downsampling strategy to sample a subset called key-events. Then, we employ a graph-based network to embed the *irregular* spatio-temporal information of key-events into a high-dimensional feature space, and the resulting embeddings are utilized to predict their target likelihoods via semantic-driven Siamese-matching. Besides, we also propose motion-aware target likelihood prediction, which learns the motion flow to *back-trace* the potential initial positions of key-events and measures them with the previous proposal. Finally, we obtain the bounding box by adaptively fusing the two intermediate ones separately regressed from the weighted embeddings of key-events by the two types of predicted target likelihoods. Extensive experiments on both synthetic and real event datasets demonstrate the superiority of the proposed framework over state-of-the-art methods in terms of both the tracking accuracy and speed. The code is publicly available at https://github.com/ZHU-Zhiyu/Event-tracking.

## 1 Introduction

Event cameras, the bio-inspired silicon retinas, have been widely adopted in computer vision and robotics communities [57, 37, 18, 31, 28, 47]. Different from traditional RGB cameras, which synchronously record the light intensity of each pixel, event cameras [13] only sense the changes in dynamic scenes, i.e., if an event senor captures an intensity increment (resp. decrement) larger than a threshold, it would trigger an event, composed of the corresponding pixel location, the timestamp, and the polarity indicating brightness increase or decrease. With the concise information flow, event cameras enjoy numerous advantages, such as high temporal resolution, low latency, high dynamic range, and low power consumption. Moreover, recent works show that even without being associated with RGB images, only event data could also realize various vision tasks, such as depth estimation [14], semantic segmentation [1], and object detection [32].

---

*Corresponding author

36th Conference on Neural Information Processing Systems (NeurIPS 2022).

Notably, owing to the above-mentioned advantages, event cameras are desired to be a promising object tracking solution [39, 18, 50], aiming to estimate the position of a specified target object. However, the unusual data structure of the raw event data poses great challenges in designing tracking algorithms. Facing such challenges, the existing non-learning-based methods usually build their models based on various representations, such as event frame [40, 17, 29], time surface [33, 55, 7], and event clouds [2, 32, 45]. In view of the great success of deep learning techniques in various fields, especially RGB data-based object tracking, recently several deep learning-based methods for event data-based object tracking have been naturally proposed [51, 47]. Generally, they quantize and accumulate raw event data along the temporal dimension into event frames/images to adapt to mature RGB data-based object tracking paradigms. However, such a manner drops the high temporal resolution and sparse characteristics of raw event data to some extent, thus compromising tracking accuracy.

Instead of following the research line of the existing learning-based methods, we are interested in developing a new end-to-end learning-based paradigm that directly consumes raw event data (or event clouds), in which new algorithms/modules considering the unique features of event clouds are involved. Specifically, to cope with the irregular structure, we employ a graph-based neural network to map an event cloud into a high-dimensional feature space to embed rich spatio-temporal semantic information. Meanwhile, to process a large-scale event cloud efficiently and being aware of its non-uniform distribution characteristic, we introduce a simple yet effective density-insensitive downsampling strategy to sample a limited number of key-events, on which the feature embedding process is performed. We then link the embeddings of template and search event clouds via Siamese-matching to predict semantic-aware target likelihoods of search key-events. Besides, observing the high temporal resolution characteristic of event data naturally preserve motion trajectories, we also propose to back-trace the possible initial positions of the search key-event and measure them with the previous proposal to predict their target likelihoods. Finally, we obtain the bounding box by adaptively fusing the two intermediate ones separately regressed from the weighted embeddings of key-events by the two types of predicted target likelihoods. We conduct extensive experiments on both real and synthetic event data to demonstrate the superiority of our framework over state-of-the-art methods in terms of both the tracking *accuracy* and *speed*, and comprehensive ablation studies to validate the effectiveness of the proposed modules.

In summary, the main contributions of this paper are two-fold:

- we propose the *first* end-to-end learning-based object tracking paradigm directly handling *raw* event data (or event clouds), featured with carefully-designed modules, e.g., key-event embedding and motion-aware target likelihood prediction, to take advantage of the unique characteristics of event data; and

- The superiority performance of our paradigm certifies the great potential of designing new algorithms tailored to the special characteristics of event data to achieve object tracking. Besides, we will release our framework built from scratch to contribute to this community.

## 2 Related Work

**Deep learning on event data**. As a kind of asynchronous bio-inspired vision sensors, event cameras [13] have shown remarkable performance in various scenarios, such as quick motion and low light environments, representing a paradigm shift in visual information collection [9]. Deep learning-based frameworks for modeling event data have been proposed. For example, Phased LSTM [36] extends the LSTM unit by adding a new time gate for classification on event data. EVDL [31] proposes an event-frame representation and constructs a CNN model for prediction of a vehicle's steering angle. Then, EV-SegNet [1] introduces the first baseline for semantic segmentation with event data. The emergence of MVSEC [57] dataset has aroused wide interest in the research of optical flow [56, 58, 16] and depth estimation [58, 20]. Unlike the previous works, Rebecq *et al.* [41, 42] used simulated events to train a convolutional recurrent neural network for image reconstruction.

**RGB data-based object tracking**. Recent methods could be roughly divided into two categories: *spatio-only*-based and *spatio-temporal*-based. Most of offline Siamese-based tracking methods [3, 26, 25] fall into the first category. For example, SiamFC [3] enhances the tracking accuracy by first introducing a correlation layer as a fusion tensor. SiamRPN [26] employs a single-stage RPN [43] detector to redetect the template by comparing its features to that of the current frame.

SiamRPN++ [25] uses a spatial-aware sampling strategy to remove the influence factors, such as padding, and introduce ResNet [19] into the Siamese network-based visual trackers. However, the spatio-only methods are difficult to deal with fast-moving or similar objects in the scene.

Compared with spatio-only trackers, *spatio-temporal*-based methods [35, 23, 46, 49] additionally exploit temporal information to improve trackers' robustness. For example, MDNet [35] proposes a CNN-based multi-domain learning framework, which separates domain-independent information from domain-specific one, to capture shared representations. RT-MDNet [23] improves MDNet by employing the RoIAlign technique to extract more accurate representations of targets and candidates from a feature map. Recently, Wang *et al.* [46] converted a general object detector to achieve domain adaption and efficient online update. However, these methods are either difficult to balance performance and speed or cannot handle real-time scenarios and low light scenes well.

Although various advanced techniques have been proposed for boosting the performance of RGB data-based object tracking, the essentially different data modality requires additional efforts to extend these techniques to event data-based object tracking.

**Event data-based object tracking** has been receiving increasing attention recently, and the proposed methods could be broadly divided into two categories: optimization-based and learning-based.

For the optimization-based category, as the pioneering work, Mitrokhin *et al.* [32] proposed to approximate the 3D geometry of the event stream without feature tracking or explicit optical flow computation. Bryner *et al.* [5] presented a method to track the 6-DOF pose of an event camera in a known environment described by a photometric 3D map.

Ramesh *et al.* [39] presented the first learning-based long-term object tracking algorithm for event cameras, based on a tracking-learning-detection framework. Then, Li *et al.* [27] utilized the features by VGG-Net-16 to represent the appearance of the event-stream object to achieve tracking. Recently, Chae *et al.* [6] applied a Siamese-matching paradigm [3] to the event-based object tracking task for learning edge-aware similarity. However, the methods in [27, 6] mainly focus on applying well-developed RGB data-based tracking techniques to event data without well taking the special characteristics of event data into account.

## 3 Proposed Method

**Notation**. Denote by $\mathbf{e} := [x, y, t, p] \in \mathbb{R}^{1 \times 4}$ an event, where $x$ and $y$ are the spatial coordinates on the image plane, whose values are normalized to $[0, 1]$, $t$ is the timestamp, and $p$ is the 1-bit polarity, indicating brightness increase or decrease. Denote by $\mathbf{E} \in \mathbb{R}^{N \times 4}$ an event cloud, where $N$ events in a short period of time $[t_0, t_0 + \Delta_t]$ are randomly stacked in row-wise. Accordingly, for events in a long period of time, we represent them with an event cloud sequence of $U$ non-overlapping event clouds, i.e., $\left\{ \mathbf{E}^i \in \mathbb{R}^{N_i \times 4} \right\}_{i=1}^{U}$. Let $\mathbf{E}_t$ be a template event cloud, and $\mathbf{B}^0 \in \mathbb{R}^{1 \times 4}$ be the annotated bounding box indicating the 2-D position of the object captured by $\mathbf{E}_t \in \mathbb{R}^{N_t \times 4}$, which is parameterized by the 2-D spatial locations and the corresponding size (i.e., width and height). The objective of event-based object tracking is to predict the bounding box of a target object described by $\mathbf{B}^0$ and $\mathbf{E}_t$ from each of a given search event cloud sequence $\{\mathbf{E}_s^i \in \mathbb{R}^{N_{s_i} \times 4}\}_{i=1}^{U_s}$.

**Overview**. In contrast to existing learning-based methods usually re-organizing raw event data with the frame-based representation, which compromises the high temporal resolution characteristic to some extent and thus limits tracking performance, we directly deal with raw event data in its intrinsic 3-D spatio-temporal space. Specifically, as shown in Fig. 1, the proposed framework processes the event clouds of $\{\mathbf{E}_s^i\}_{i=1}^{U_s}$ sequentially via the following three main steps:

*Graph-based Key-event Embedding*. This module aims to project an event cloud into a high-dimensional feature space to embed rich spatio-temporal semantic information (see Sec. 3.1). However, the irregular structure underlying event clouds, caused by the sparse and asynchronous sensing manner, makes it intractable to learn the embeddings efficiently and effectively. To this end, we employ a GNN, which has demonstrated the effectiveness in modeling irregularly distributed signals, to embed input raw event data. Moreover, to process large-scale event clouds efficiently and being aware of their noisy and highly non-uniform characteristics, we propose a simple yet effective downsampling strategy, capable of sampling a few key-events evenly and robustly from an input

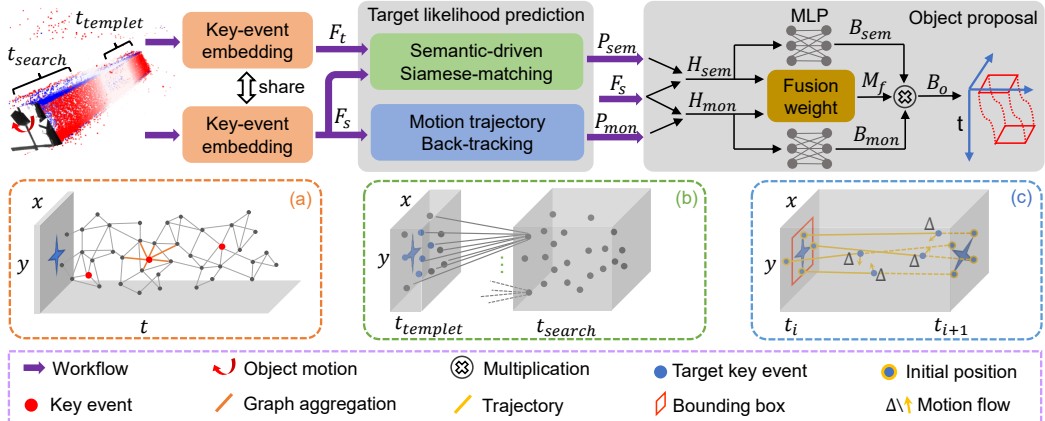

Figure 1: Flowchart of the proposed end-to-end learning-based framework for tracking objects in event clouds. It mainly consists of three steps: GNN-based key-event embedding, semantic matching and motion-aware target likelihood prediction and confidence-based object proposal. Sub-figs. (a), (b), and (c) illustrate the detailed processes of the first two steps, marked with corresponding color .

event cloud. Accordingly, we perform the tracking process on key-events instead of the whole event cloud.

*Semantic and Motion-aware Target Likelihood Prediction.* This module, a dual-path architecture, aims to predict the possibility of a key-event of the search event cloud belonging to the target object, i.e., the *target likelihood*. Basically, from the visual recognition perspective, we propose a semantic-driven Siamese-matching module, in which the similarity of each search key-event to the template key-events is evaluated, based on their embeddings (see Sec. 3.2). Besides, observing that the event cloud inherently preserves the motion trajectories of events, we also propose a motion-aware target likelihood prediction module, in which the initial position of each key-event is back-traced and compared with the previous proposal (see Sec. 3.3). We expect these two paths designed from different perspectives will compensate each other for robust and accurate target likelihood prediction.

*Confidence-based Object Proposal.* In this module, we first regress two intermediate bounding boxes separately from the embeddings of key-events weighted by the two types of predicted target likelihoods and then fuse them adaptively under the guidance of the learned confidence scores, generating the final bounding box (see Sec. 3.4).

## 3.1 Graph-based Key-event Embedding

We first sample a limited number of key-events via a simple yet effective density-insensitive down-sampling strategy, then embed them into a high-dimensional feature space via a GNN.

**Key-event sampling**. As shown in Fig. 2 (a), to sample $n_k$ key-events from a typical event cloud, we first define a 2D regular grid that contains $n_k$ grid points and has the same scale as the event cloud in the spatial domain. Then, for each of the grid points, we find the spatially closest event in the Euclidean distance sense, which will be thought of as a key-event. Such a simple downsampling strategy could spread the receptive field of the neural network in a more spatially uniform manner. We demonstrate its superiority against widely-used random sampling [21] in Sec. 4.3.

**GNN-based spatio-temporal embedding**. We utilize the graph convolutional layers to encode the irregular 3-D spatio-temporal information of an event. Specifically, as shown in Fig. 2 (a), for each key-event, we first figure out its nearest neighbours, then learn the high-dimensional spatio-temporal embeddings of such a set of events via feeding the explicitly encoding[2] of event clouds to multi-layer perceptrons (MLPs). Meanwhile, adaptive pooling weights are also learned to aggregate neighbouring information in an unordered manner, finally producing embeddings $\mathbf{F}_t \in \mathbb{R}^{n_t \times C}$ and $\mathbf{F}_s \in \mathbb{R}^{n_s \times C}$ for $\mathbf{E}_t$ and $\mathbf{E}_s^i$ containing $n_t$ and $n_s$ key-events, respectively, where $C$ is the number of channels. We refer readers to *Appendix* for more details.

---

[2]The encoding refers to the concatenation of the central event point $\mathbf{e}_c$, its neighbour $\mathbf{e}_n$, the difference $\mathbf{e}_c - \mathbf{e}_n$, and the Euclidean distance $\|\mathbf{e}_c - \mathbf{e}_n\|_2$.

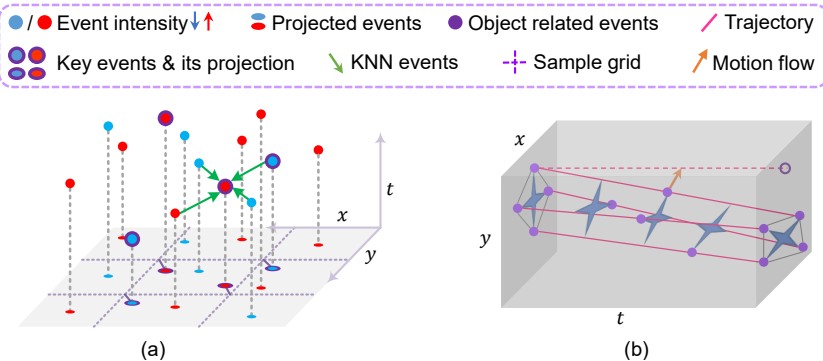

Figure 2: Illustration of (a) the density-insensitive event downsampling strategy and local spatio-temporal aggregation pattern of the graph neural network, and (b) the key-event back-tracing process used in motion-aware target likelihood prediction.

### 3.2 Target Likelihood Prediction via Semantic-driven Siamese-matching

Let $p_{sem}^j \in [0,1]$ be the target likelihood of the $j$-th search key-event, and $\mathbf{P}_{sem} := [p_{sem}^1; \cdots; p_{sem}^j; \cdots; p_{sem}^{n_s}] \in \mathbb{R}^{n_s \times 1}$. Encouraged by the learnable combination pattern of transformer, which has shown its great potential in sparse point cloud processing [52], we first apply swin-transformer [30] on the embeddings $\mathbf{F}_t$ and $\mathbf{F}_s$, which respectively encode the semantics of the key-events of $\mathbf{E}_t$ and $\mathbf{E}_s^i$, to equip them with abstract and distinguishable information, generating $\mathbf{S}_t \in \mathbb{R}^{n_t \times C_o}$ and $\mathbf{S}_s \in \mathbb{R}^{n_s \times C_o}$ ($C_o > C$). To link $\mathbf{E}_t$ and $\mathbf{E}_s^i$, we then design a cross-attention layer, which adaptively projects $\mathbf{S}_t$ onto $\mathbf{S}_s$:

$$\mathbf{S}_{ts} = \mathbf{M}_a \times \mathbf{S}_s, \quad \mathbf{M}_a = \mathcal{G}(\mathbf{S}_t, \mathbf{S}_s), \tag{1}$$

where the associated matrix $\mathbf{M}_a \in \mathbb{R}^{n_t \times n_s}$ is learned via another GNN $\mathcal{G}(\cdot, \cdot)$, and $\mathbf{S}_{ts} \in \mathbb{R}^{n_t \times C_o}$ denotes the projected embeddings blending the information of both $\mathbf{E}_t$ and $\mathbf{E}_s^i$. Finally, we feed the concatenation of $\mathbf{S}_{ts}$ and $\mathbf{S}_s$ into another MLP to regress $\mathbf{P}_{sem}$. We refer readers to *Appendix* for the detailed architecture.

### 3.3 Target Likelihood Prediction via Motion Trajectory Back-tracking

In this module, we leverage the inherent motion trajectories of event data [58, 34, 40] to predict the target likelihood of each key-event of $\mathbf{E}_s^i$. The basic idea is linking the current search key-events to the previous target proposal to distinguish them. Specifically, we first back-trace the initial position of each key-event from its motion trajectory in 3D spatio-temporal space. Considering that trajectories are usually continuous, the former event points should have relatively smaller moving offsets. To explicitly model such a characteristic, we apply MLPs to the embeddings of search key-events to learn their motion velocities $\mathbf{V} \in \mathbb{R}^{n_s \times 2}$, which are further used to calculate the motion flow $\mathbf{M} \in \mathbb{R}^{n_s \times 2}$, i.e.,

$$\mathbf{M} = \mathbf{V} \times \mathbf{T}/\max(\mathbf{T}), \quad \mathbf{V} = \texttt{MLP}(\mathbf{F}_s), \tag{2}$$

where $\mathbf{T} \in \mathbb{R}^{n_s \times 1}$ records the timestamps of the search key-events, and $\max(\cdot)$ is the operation selecting the maximum element of the input. With the motion flow and the positions of the search key-events, denoted as $\mathbf{K} \in \mathbb{R}^{n_s \times 2}$, we can localize their initial positions, denoted by $\mathbf{K}_{ini} \in \mathbb{R}^{n_s \times 2}$, as $\mathbf{K}_{ini} = \mathbf{M} + \mathbf{K}$. Finally, we predict the motion-aware target likelihood of each search key-event, denoted as $p_{mon}^j \in [0, 1]$ (accordingly, $\mathbf{P}_{mon} := [p_{mon}^1; \cdots; p_{mon}^j; \cdots; p_{mon}^{n_s}] \in \mathbb{R}^{n_s \times 1}$), from $\mathbf{K}_{ini}$ together with the previous bounding box and a normalized distance $\left[\frac{(x-x_b)}{w_b}\right]^2 + \left[\frac{(y-y_b)}{h_b}\right]^2$ independently via MLPs, where $x_b$, and $y_b$ are the spatial coordinates of the box center, and $w_b$ and $h_b$ are the width and height of the box, respectively. We refer readers to *Appendix* for the detailed architecture.

### 3.4 Confidence-based Object Proposal

This module regresses a bounding box from the predicted two kinds of predicted target likelihoods $\mathbf{P}_{sem}$ and $\mathbf{P}_{mon}$, as well as the embeddings $\mathbf{F}_s$ encoding the rich spatio-temporal semantic informa-

tion of $\mathbf{E}_s^i$. Specifically, taking the semantic-aware branch as an example, we formulate the bounding box regression as

$$\mathbf{B}_{sem} = \mathtt{MLP}(\mathbf{H}_{sem}), \ \mathbf{H}_{sem} = \mathbf{M}_{p_{sem}} \times (\mathbf{F}_s \odot \mathtt{D}(\mathbf{P}_{sem})), \ \mathbf{M}_{p_{sem}} = \mathcal{S}(\mathtt{MLP}(\mathbf{F}_s \odot \mathtt{D}(\mathbf{P}_{sem}))), \ (3)$$

where $\mathbf{B}_{sem} \in \mathbb{R}^{1 \times 4}$ denotes the intermediate bounding box, based on $\mathbf{P}_{sem}$, $\mathbf{H}_{sem} \in \mathbb{R}^{1 \times c}$ encodes the embeddings after adaptive pooling, $\mathbf{M}_{p_{sem}} \in \mathbb{R}^{1 \times n_k}$ is the pooling weights, $\odot$ denotes the hadamard product, $\mathtt{D}(\cdot)$ is the duplication operation, and $\mathcal{S}(\cdot)$ stands for the softmax function. We also apply the same procedures to the motion-aware branch, leading to another intermediate bounding box $\mathbf{B}_{mon}$. Finally, we fuse the two intermediate bounding boxes under the guidance of learned confidence scores, i.e.,

$$\mathbf{B}_o = \mathbf{M}_f \times [\mathbf{B}_{sem}; \mathbf{B}_{mon}], \ \ \mathbf{M}_f = \mathcal{S}\{[\mathtt{MLP}(\mathbf{H}_{sem}); \mathtt{MLP}(\mathbf{H}_{mon})]\}, \tag{4}$$

where $\mathbf{B}_o$ denotes the final output bounding box, and $\mathbf{M}_f \in \mathbb{R}^{1 \times 2}$ refers to the learned confidence scores. We refer readers to *Appendix* for the detailed architecture.

### 3.5 Training & Online Tracking

**Loss function**. To train the proposed framework end-to-end, we design the following loss function:

$$\mathcal{L}(\mathbf{B}_g, \hat{\mathbf{B}}, \mathbf{P}_g, \hat{\mathbf{P}}) = \sum_{\mathbf{P}_k \in \hat{\mathbf{P}}} \mathcal{L}_{cls}(\mathbf{P}_g, \mathbf{P}_k) + \alpha \sum_{\mathbf{B}_k \in \hat{\mathbf{B}}} (\mathcal{L}_{reg}(\mathbf{B}_g, \mathbf{B}_k) + \mathcal{L}_{giou}(\mathbf{B}_g, \mathbf{B}_k)) + \beta \mathcal{L}_{CD}, \ (5)$$

where $\hat{\mathbf{B}} := \{\mathbf{B}_o, \mathbf{B}_{sem}, \mathbf{B}_{mon}\}$, $\hat{\mathbf{P}} := \{\mathbf{P}_{sem}, \mathbf{P}_{mon}\}$, $\mathbf{B}_g$ and $\mathbf{P}_g$ are the ground-truth bounding box and target likelihood, respectively, $\mathcal{L}_{cls}(\cdot, \cdot)$ is the binary cross-entropy loss to supervise the target likelihood prediction, $\mathcal{L}_{reg}(\cdot, \cdot)$ refers to the $\mathcal{L}_1$ loss to supervise the regression of the bounding box, $\mathcal{L}_{giou}(\cdot, \cdot)$ is the generalized intersection over union (GIOU) loss [44], and $\mathcal{L}_{CD}$ computes the Chamfer distance between the back-traced key-events and their potential locations to supervise motion flow estimation, defined as

$$\mathcal{L}_{CD} = \frac{1}{n_s} \sum_{\mathbf{k}_c \in \mathbf{K}_c} \min_{\mathbf{k}_i \in \mathbf{K}_{ini}} \mathcal{M}(\|\mathbf{k}_i - \mathbf{k}_c\|, \gamma) + \frac{1}{n_c} \sum_{\mathbf{k}_i \in \mathbf{K}_{ini}} \min_{\mathbf{k}_c \in \mathbf{K}_c} \mathcal{M}(\|\mathbf{k}_i - \mathbf{k}_c\|, \gamma), \tag{6}$$

where $\mathbf{K}_{ini} \in \mathbb{R}^{n_c \times 2}$ denotes the back-projected key-events. We clip the event clouds within very tiny time interval $\mathbf{K}_c \in \mathbb{R}^{n_c \times 2}$, $t_0 \leq \mathbf{t}_c \leq t_0 + \mu \Delta_t$, as the potential locations of $\mathbf{K}_{ini}$. Due to the inevitable noise events, we truncate the accidental errors by a threshold $\gamma$ via a minimization function $\mathcal{M}(\cdot)$. we empirically set the hyper-parameters $\alpha$, $\beta$, $\mu$, and $\gamma$ to $1e^0$, $1e^1$, $2e^{-1}$, and $1e^{-1}$, respectively.

**Robust training strategy**. In addition to $\mathbf{E}_t$ and $\mathbf{E}_s^i$, we also have to feed the proposed framework with the previous bounding boxes. Accordingly, the motion-aware target likelihood prediction may highly rely on the accuracy of previous predictions. To mitigate the potential accumulated errors and improve the robustness of our framework, we disturb the input bounding boxes with random Gaussian noises during training.

**Template updating strategy for online tracking**. Considering that the appearance of an object may undergo a large variance during a tracking process, we also introduce a strategy to update template key-events. Specifically, we first partition the embeddings of key-events of $\mathbf{E}_t$, i.e., $\mathbf{S}_t$, into two sets equally: the constant set $\mathbf{S}_{tc}$ and the variant set $\mathbf{S}_{tv}$. Then, after each search procedure, we stack in the embeddings of the key-events of $\mathbf{E}_s^i$ (i.e., $\mathbf{S}_s$ from) with a high target likelihood, i.e., $p > \tau$, where $\tau$ is empirically set to $9e^{-1}$, to $\mathbf{S}_{tv}$ and randomly pop out embeddings from $\mathbf{S}_{tv}$ to keep the total size of $\mathbf{S}_t$ unchanged.

## 4 Experiments

### 4.1 Experiment Settings

**Datasets**. We evaluated the proposed method with both real and synthetic event datasets. The real event dataset *FE108* (MIT) [51] provides event data of spatial dimensions $346 \times 260$, dynamic range 120dB, and minimum latency time 20 $\mu s$, captured by the DAVIS 346 camera. The ground-truth bounding boxes were captured by the Vicon motion capture system. *FE108* contains 108 sequences

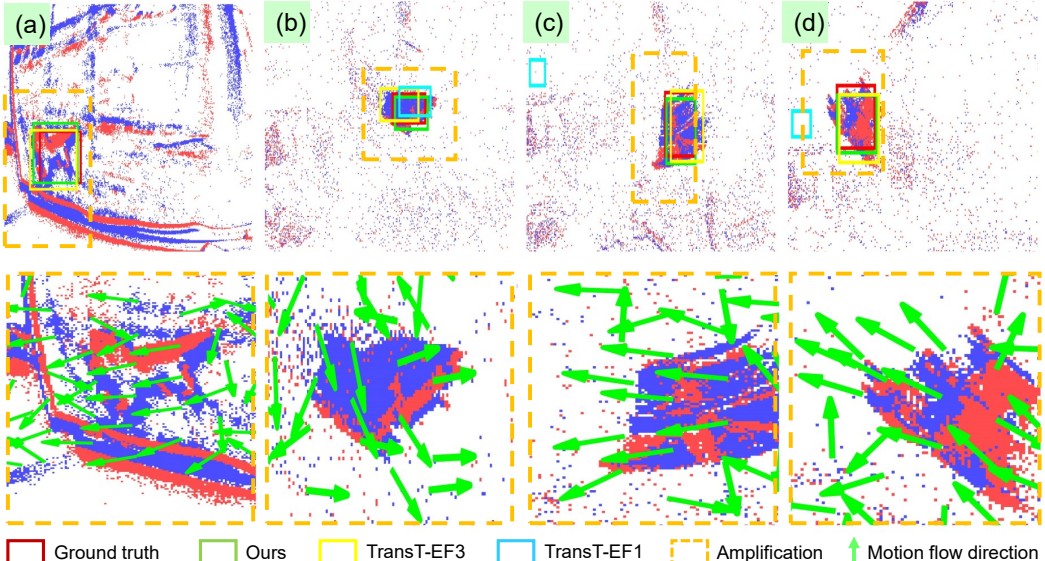

| Ground truth | Ours | TransT-EF3 | TransT-EF1 | Amplification | Motion flow direction |

Figure 3: *Upper row:* visual comparison of the tracking results of different methods on four event clouds randomly selected from the FE108 dataset. *Bottom row:* a local zoom-in view of the dashed box in corresponding upper sub-figures. The green arrows start from the spatial locations of key-events and point to the direction of motion flow.

of total length 1.5 hours, recording the motion of 21 different types of objects under four challenging scenarios. There are 59K event clouds for validation and 140K event clouds for training. Considering that most objects in *FE108* are rigid, which may be insufficient to validate the performance on non-rigid objects, we also constructed a new synthetic dataset *Event LaSOT (Evt-LaSOT)*, simulated from the widely-used RGB data-based object tracking dataset *LaSOT* (Apache-2.0) [12] via Vid2E[15]. *Evt-LaSOT* consists of eight sequences[3] randomly drawn from LaSOT , and there are 18K event clouds in total.

**Compared methods**. We compared the proposed method with SOTA RGB data-based object tracking algorithms, including SiamRPN [26], ATOM [10], DiMP [4], SiamFC++ [48], and PrDiMP [11], and the event data-based branch of FENet [51]. Moreover, we modified two SOTA RGB data-based object tracking methods, i.e., PrDiMP [8] and TransT [8], to adapt them to the event frame representation [51], leading to two strong baselines, namely E-PrDiMP and E-TransT. For fair comparisons, we trained all methods with the training set of *FE108* and evaluated them on the testing set of *FE108* and *Evt-LaSOT*.

**Implementation details**. To train our framework in a batch manner, we sampled 10K event points for each event cloud. We set the value of $\Delta_t$ as the interval of annotations, i.e., $25 \sim 50ms$ for the FE108 dataset according to its 20/40 FPS annotations. We implemented the framework with Pytorch on Ubuntu 18.04 and trained it with $4 \times$ NVIDIA-3090 and a batch size of 22 samples in each GPU , about 300K iterations. We adopted the Adam optimizer [24] with the weight decay of $1e^{-5}$ and the learning rate of $1e^{-5}$ (resp. $1e^{-4}$) for the backbone (resp. the two target likelihood prediction and object proposal modules). We set the spatial search region of the current event cloud as $1.2\times$ the size of the previous proposal.

### 4.2 Results

**Results on FE108**. Table 1 and Fig. 4 (a) and (b) show the quantitative comparisons of different methods on *FE108*, where it can be seen that the event-based methods generally surpass the RGB data-based methods under the LL and HDR scenarios, demonstrating the advantage of event data. Besides, the significant improvement of E-TansT with the input changed from EF-1 to EF-3 demonstrates that the high temporal resolution characteristic of event data is of importance. Our method achieves the

---

[3]The eight sequences are "bird-15", "bird-3", "crab-6", "crab-18", "cat-20", "cat-3", "crocodile-3", and "bear-4".

Table 1: Quantitative comparison on the *FE108* dataset in terms of four commonly-used metrics, i.e., representative success rate (RSR), representative precision rate (RPR), and overlap percision (OP) with the threshold equal to 0.5 ($OP_{0.50}$) and 0.75 ($OP_{0.75}$). For all metrics, the larger, the better. There are four challenging scenarios, i.e., low-light (LL), high dynamic range (HDR), fast motion with and without motion blur on image frame (FWB and FNB), and all testing dataset (ALL). "Img" (resp. "EF$-n$") indicates that the input of the methods is the RGB image/video (resp. the event frame/image), where $n$ refers to the number of channels or the temporal resolution. "RawE" stands for raw event data (or event clouds). The best and second best results of event-based methods are highlighted in red and green, respectively.

| Metrics | | ATOM [10] | DiMP [4] | SiamFC++ [48] | PrDiMP [11] | E-PrDiMP [11] | E-TransT [8] | E-TransT [8] | FENet [51] | Ours |
|---|---|---|---|---|---|---|---|---|---|---|
| Input | | Img | Img | Img | Img | EF-3 | EF-1 | EF-3 | EF-3 | RawE |
| HDR | RSR | 32.3 | 41.8 | 15.3 | 44.3 | 52.8 | 42.4 | 50.7 | 50.0 | 50.9 |
| | $OP_{0.50}$ | 36.9 | 50.0 | 15.0 | 52.8 | 62.5 | 51.4 | 60.8 | 57.8 | 60.7 |
| | $OP_{0.75}$ | 11.8 | 17.9 | 1.3 | 19.6 | 23.8 | 17.5 | 23.0 | 18.8 | 17.2 |
| | RPR | 55.2 | 62.7 | 25.2 | 66.3 | 84.6 | 70.5 | 84.2 | 77.4 | 81.5 |
| LL | RSR | 38.6 | 45.6 | 13.4 | 44.6 | 58.0 | 46.8 | 58.8 | 56.6 | 60.6 |
| | $OP_{0.50}$ | 44.3 | 52.8 | 8.7 | 48.2 | 67.8 | 55.9 | 72.2 | 69.1 | 75.0 |
| | $OP_{0.75}$ | 13.2 | 11.2 | 0.8 | 8.9 | 28.7 | 25.0 | 31.4 | 22.9 | 30.8 |
| | RPR | 59.6 | 69.5 | 15.3 | 69.5 | 86.2 | 72.6 | 92.2 | 86.1 | 90.1 |
| FWB | RSR | 44.1 | 69.4 | 28.6 | 67.0 | 43.3 | 31.6 | 46.8 | 59.3 | 60.8 |
| | $OP_{0.50}$ | 50.9 | 94.7 | 36.3 | 89.9 | 52.9 | 34.3 | 57.9 | 74.2 | 76.3 |
| | $OP_{0.75}$ | 16.7 | 37.1 | 6.0 | 33.6 | 19.6 | 0.09 | 28.4 | 32.1 | 29.8 |
| | RPR | 88.6 | 99.7 | 48.2 | 99.7 | 81.6 | 77.0 | 87.9 | 84.9 | 90.6 |
| FNB | RSR | 57.1 | 60.5 | 36.8 | 60.6 | 47.7 | 30.7 | 51.8 | 50.2 | 54.3 |
| | $OP_{0.50}$ | 68.2 | 75.6 | 42.7 | 75.8 | 51.0 | 35.7 | 60.2 | 56.3 | 61.8 |
| | $OP_{0.75}$ | 32.1 | 29.3 | 7.4 | 29.7 | 18.9 | 12.9 | 24.8 | 19.0 | 18.7 |
| | RPR | 87.9 | 93.2 | 63.1 | 93.3 | 80.0 | 52.9 | 85.8 | 75.5 | 87.0 |
| ALL | RSR | 40.6 | 52.6 | 23.8 | 53.0 | 52.0 | 39.1 | 52.4 | 52.3 | 54.9 |
| | $OP_{0.50}$ | 47.8 | 65.4 | 26.0 | 65.0 | 60.9 | 46.2 | 62.2 | 61.4 | 65.8 |
| | $OP_{0.75}$ | 17.3 | 23.4 | 3.9 | 23.3 | 22.6 | 16.6 | 21.0 | 19.8 | 21.4 |
| | RPR | 67.0 | 79.1 | 39.1 | 80.5 | 84.6 | 67.2 | 87.0 | 80.0 | 85.9 |

best or second best performance in terms of different metrics under most scenarios, and especially, it significantly exceeds the compared methods under the fast motion scenario. The reason may be that the compared methods are disturbed by the large motion offset in the accumulated event frames, as shown in Fig. 3 (b). Although the proposed method is slightly worse than or comparable to the compared event frame-based methods under very rare scenarios, we want to notice that those methods benefit a lot from the RGB data-based tracking, a long-standing field featured with great efforts devoted and mature techniques, while our new paradigm is still plain to some extent, and it is highly expected that its performance raises gradually with more advanced modules incorporated in the future. **Results on Evt-LaSOT**. Since these synthetic event data from RGB data do not have high temporal resolution and are driven from estimated optical-flow, which may not correctly reflect the real movement of an object, for the sake of fairness, we only compared the proposed method with event-based methods on this dataset. As shown in Table 2 and Figs. 4 (c) and (d), the proposed method trained with real event data from *FE108* is well generalized on *Evt-LaSOT* and outperforms all the compared methods to a significant extent in terms of all the three metrics, which is credited to the new paradigm of directly modeling raw event data. Besides, since our method directly consumes raw event data without pre-processing required by existing event frame/image-based methods, it is more efficient, reflected by the higher FPS. We also refer readers to the *video demo* contained in *Supplementary Material* for the visual comparison of different methods.

**Comparison with 3D point cloud-based trackers.** Recently 3D point cloud-based object tracking achieves remarkable growth, which aims to predict the location of a target object from a 3D point cloud (PC) sequence [38, 54, 22, 53]. Although event clouds could be thought of as 3D point clouds, they are still different. Specifically, 3D point clouds by the LiDAR sensor record the geometric information of 3D entities, which are generally collected by measuring the reflection time of the emitted pulses of infrared light hitting nearby objects/scenes. Only the part of the object/scene visible to the LiDAR sensor will be perceived (one emitted pulse will generate at most one 3D point). Thus, the captured 3D points are distributed on the surface of the 3D entity. However, even data are acquired by measuring the pixel intensity variations in a short time period, and for a typical pixel, there may

Table 2: Quantitative comparison on the *Evt-LaSOT* dataset. FPS refers to the number of times of prediction per second processed during inference. We refer readers to *Appendix* for the metric details. The best results are highlighted in bold.

| Metrics | | E-PrDiMP [11] | E-TransT [8] | FENet [51] | Ours |
|---|---|---|---|---|---|
| FPS | | 15.7 | 18.7 | 17.2 | **25.2** |
| Evt-LaSOT | RSR | 27.9 | 30.3 | 28.7 | **32.3** |
| | $OP_{0.50}$ | 12.3 | 17.9 | 15.4 | **22.1** |
| | $RPR_{0.075}$ | 25.4 | 30.1 | 26.9 | **35.9** |

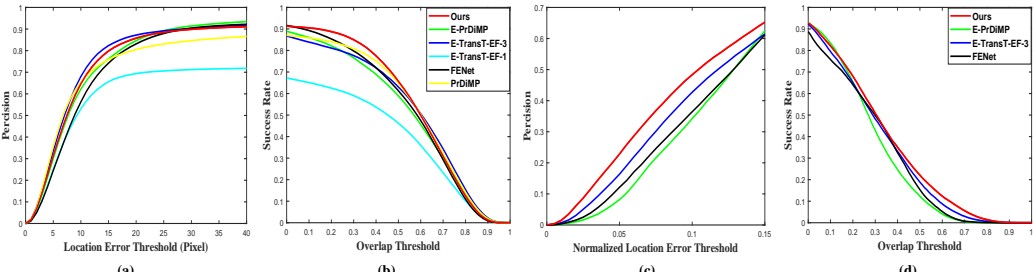

Figure 4: The precision and success plots of different methods, where sub-figure (a) and (b) are shown for *FE108*, (c) and (d) for *Evt-LaSOT*.

be multiple events (i.e., the events have the same $x$ and $y$ dimensions but different $t$ dimensions). Thus, the events are distributed within a 3D cube. Such a data distribution difference may make it inappropriate to apply 3D point cloud-based trackers to event data. Here, we also conducted experiments to compare the proposed method with SOTA 3D point cloud-based trackers on FE108 dataset. As listed in Table 3, the proposed method exceeds the 3D point cloud-based trackers [38, 54] to a large extent, i.e., 7.4% and 15.5% improvements in terms of the success rate and precision respectively, demonstrating the demand of designing event-specialized tracking algorithms.

### 4.3 Ablation Studies

**Event downsampling strategy**. We constructed a baseline by replacing the proposed downsampling strategy with random sampling. By comparing rows 4) and 6) of Table 4, it can be seen that without using the proposed downsampling strategy, the values of RSR and RPR decrease by 4.8% and 8.4%, respectively, validating the effectiveness of this strategy. Moreover, such a downsampling process can accelerate the network convergence during training.

**Motion-aware target likelihood prediction & Training strategy**. To validate the effectiveness of this module, we constructed a baselines without using it. As shown in rows 1), 2), and 3) of Table 4, it can be seen that this module augmented by the training strategy achieves 2.7% gains in terms of RSR. Besides, without using the training strategy, this module even harms the accuracy, validating the effectiveness of the training strategy on mitigating the accumulated errors.

**Flow estimation module.** By comparing rows 3) and 6) of Table 4, it can be seen that this module improves the value of RSR by 1%, validating its effectiveness. Besides, we also visualized the flow

Table 3: Comparison of the proposed method and 3D point cloud-based trackers on the FE108 dataset. We trained the 3D point cloud-based trackers with the same experiment settings as ours and also tried our best to adapt those methods to event data for optimal performance.

| method | RSR | $OP_{0.50}$ | $OP_{0.75}$ | RPR |
|---|---|---|---|---|
| P2B [38] | 36.1 | 35.4 | 4.7 | 40.5 |
| PTTR [54] | 47.5 | 49.6 | 11.5 | 70.2 |
| Ours | 54.9 | 65.8 | 21.4 | 85.9 |

Table 4: Quantitative results of the ablation studies of the proposed method. "×" (resp. ✓) means the corresponding module is unused (resp. used). (a) Downsampling strategy (b) Injecting noise during the training phase (c) Flow estimation (d) Motion-aware target likelihood prediction (e) online template updating strategy (f) Semantic-aware target likelihood prediction. Note that (b) and (c) are applicable only if (d) is used.

| | (a) | (b) | (c) | (d) | (e) | (f) | RSR | $OP_{0.50}$ | $OP_{0.75}$ | RPR |
|---|---|---|---|---|---|---|---|---|---|---|
| 1) | ✓ | × | × | × | ✓ | ✓ | 52.2 | 61.5 | 14.2 | 85.0 |
| 2) | ✓ | × | × | ✓ | ✓ | ✓ | 51.7 | 61.2 | 14.5 | 83.7 |
| 3) | ✓ | ✓ | × | ✓ | ✓ | ✓ | 53.9 | 64.0 | 19.9 | 84.9 |
| 4) | × | ✓ | ✓ | ✓ | ✓ | ✓ | 50.1 | 59.5 | 19.3 | 77.5 |
| 5) | ✓ | ✓ | ✓ | ✓ | × | ✓ | 54.5 | 65.2 | 21.0 | 85.7 |
| 6) | ✓ | ✓ | ✓ | ✓ | ✓ | ✓ | 54.9 | 65.8 | 21.4 | 85.9 |

of key-events Fig. 3, where it can be seen that although there are some noise events, the predicted flow could correctly present the motion of both the background and target object.

**Online template updating strategy**. The effectiveness of this strategy is validated by comparing rows 5) and 6) of Table 4, where it can be see that the value of RSR improves by $0.4\%$ when including this strategy.

### 4.4 Discussions

**Limitations:** although the proposed method demonstrates the great potential of raw event data-based object tracking, there are still some promising directions that are worth studying to further boost tracking performance. First, noisy events are inevitable, as shown in Fig. 3, which may disturb the target likelihood prediction; hence, a typical event data denoising algorithm could be applied as pre-processing to remove the interference. Second, in addition to the prior knowledge of the continuous motion trajectory preserved in event data, the *velocity* information of an object hidden in event data, which may also be continuous, could be taken into account. Third, for the long-term object tracking, it is necessary to investigate more delicate template updating strategies. Finally, according to RGB data-based object tracking, the powerful pre-trained backbones are essential for improving tracking performance, and thus, it is highly desired to introduce pre-trained backbone on event data into event-based object tracking.

## 5 Conclusion

We have presented the first end-to-end learning-based object tracking paradigm on raw event data. Owing our critical modeling of the tracking process for exploring the special characteristics of raw event data, including the delicate downsampling method, the motion-aware module, and the noise injecting training strategy, the proposed framework achieves competitive performance and efficiency in comparison with state-of-the-art methods.

Last but not least, despite that we demonstrate the great potential of designing new algorithms tailored to the special characteristics of event data to achieve object tracking, as a beginning of this research line, our framework is still relatively plain, compared to the well-developed RGB data-based object tracking, and it could be expected that the performance will improve gradually with more efforts and advanced techniques incorporated in the future.

**Acknowledgements.** Funding in direct support of this work: Hong Kong Research Grants Council under Grant 11202320, Grant 11219422, and Grant 11218121.

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

Table 5: Quantitative comparison on the *FE108* dataset, with different random seeds.

| Seeds | RSR | $OP_{0.5}$ | $OP_{0.75}$ | RPR |
|-------|------|------|------|------|
| 50 | 54.9 | 65.8 | 21.4 | 85.9 |
| 100 | 54.9 | 65.9 | 21.6 | 85.7 |
| 200 | 54.9 | 65.8 | 21.4 | 85.8 |

   (d) Did you include the total amount of compute and the type of resources used (e.g., type of GPUs, internal cluster, or cloud provider)? [Yes] Please refer to Sec. 4.1.

4. If you are using existing assets (e.g., code, data, models) or curating/releasing new assets...
   (a) If your work uses existing assets, did you cite the creators? [Yes] We have correctly cited the FE108 dataset in Sec. 4.1.
   (b) Did you mention the license of the assets? [Yes] Please refer to Sec. 4.1.

(c) Did you include any new assets either in the supplemental material or as a URL? [Yes] We have included the source code and pre-trained model in the supplementary material. And we will also release the simulated Evt-LaSOT.

(d) Did you discuss whether and how consent was obtained from people whose data you're using/curating? [Yes] Since there is no random process for those image-based methods during inference. Their result is constant.

(e) Did you discuss whether the data you are using/curating contains personally identifiable information or offensive content? [Yes] There is no person related data in our dataset.

5. If you used crowdsourcing or conducted research with human subjects

(a) Did you include the full text of instructions given to participants and screenshots, if applicable? [N/A] No research with human subjects.

(b) Did you describe any potential participant risks, with links to Institutional Review Board (IRB) approvals, if applicable? [N/A]

(c) Did you include the estimated hourly wage paid to participants and the total amount spent on participant compensation? [N/A]

