# OpenReview forum: "Learning Graph-embedded Key-event Back-tracing for Object Tracking in Event Clouds"
_NeurIPS.cc/2022/Conference — NeurIPS 2022 Accept_

### Official Review · Reviewer_kchd · 2022-07-09

**Rating:** 4
**Confidence:** 4
**Soundness:** 2 fair
**Presentation:** 2 fair
**Contribution:** 2 fair

**Summary:**

Unlike existing methods that usually reorganize raw event data to solve the unusual data structure problem, this work develops new designs tailored to the unique data structure to realize object tracking. To this end, the authors construct a new end-to-end learning-based paradigm that directly consumes event clouds. The proposed method featured key-event embedding and motion-aware target likelihood prediction to take advantage of the unique characteristics of event data. Extensive experiments on both synthetic and real event datasets demonstrate the excellent performance of the proposed method in terms of both tracking accuracy and speed.

**Questions:**

1.	What is the correspondence between subfigures in the upper and lower lines of Figure 3? Can you further explain how to understand the green arrow in the second row? (Number of arrows, consistency of direction, the thickness of arrows, etc.)
2.	In Section 4.1, what is the sequence length of the datasets used in this experiment (mean frame, min frame, max frame)? Does the sequence length have a greater impact on the experimental results?
3.	As far as I know, LaSOT contains 1400 sequences, and the sequence length varies from 1K to 10K frames, so how does the Evt-LaSOT construct in this article filter data from the original LaSOT? Are there specific standards? Is the filtered sequence have the same length as the original sequence?
4.	Table 2 presents the proposed algorithm has a higher FPS on Evt-LaSOT. The authors point out that the main reason is this method directly consumes raw event data without pre-processing. Is there any other reason besides this? Can you design some ablation experiments to analyze the processing speed?
5.	The video demo's main screen intuitively shows this method's effect, but what do the three sub-screens on the right refer to? Please explain the specific meaning of each sub-screen.


**Limitations:**

The authors discuss the public privacy issues in the additional material and argue that the event-based vision puts a lower threat to public privacy than RGB image. However, I want to know how secure and robust the event-based vision is? If we deploy the event-based scheme on self-driving systems, will it be safer than the current RGB image-based plan?

**Strengths And Weaknesses:**

This paper concentrates on the unusual data structure problem and presents the first end-to-end learning-based object tracking paradigm on raw event data. Experiments illustrate that the proposed framework achieves competitive performance and efficiency compared to SOTA methods. The article is well-structured and easy to read. However, due to space limitations, some experimental details are not fully elaborated, and the analysis of experimental results can also be further strengthened. Please refer to Questions for details.

---

> ### Author Response · Authors · 2022-08-02
> **Responses to Comments of Reviewer #kchd (Part 1)**
>
> Thanks for your time and efforts in reviewing our submission, as well as the valuable comments. The authors appreciate your recognition of our work. In the following, we will address all of your concerns point-by-point in detail.
>
> **Q1**. *What is the correspondence between subfigures in the upper and lower lines of Figure 3? Can you further explain how to understand the green arrow in the second row? (Number of arrows, consistency of direction, the thickness of arrows, etc.)*
>
> **Response**: Sorry for the confusion caused. The upper row compares the tracking accuracy of different methods on four different event cloud samples. The bottom row denotes a local zoom-in view of the dashed box in corresponding sub-figures of the upper row. The green arrows start from the spatial locations of key-events and point to the direction of motion flow. We also clarify that the thickness difference of those arrows does not mean anything and is only caused by the different ratios of zoom-in local-views. We will update this figure in the camera-ready version.
>
> **Q2**. *In Section 4.1, what is the sequence length of the datasets used in this experiment (mean frame, min frame, max frame)? Does the sequence length have a greater impact on the experimental results?*
>
> **Response**: In the following table, we list the details of the test dataset.  We also categorize  the performance of our method based on the sequence length in the following tables. Besides, the results of the second-best method Trans-T are also provided for comparisons.
>
> |  | Min length      | Max length| Avg length
> |----|----| -----|-----
> |FE108 	      | 641	       | 2400	   | 1864
> |Evt-LaSOT| 1260	       | 4499	   | 2372
>
>
> |FE108 		|Method| $RSR$| $OP_{0.50}$| $OP_{0.75}$| $RPR$
> |----|----|-----|-----|---- |-----
> |Seqs. \#Frames < 1000|Trans-T|70.4  |91.7        | 40.0        | 100.0
> |Seqs. \#Frames < 1000|Ours     |70.5  | 93.1	     | 39.8         | 100.0
> |Seqs. \#Frames > 2000|Trans-T |49.4 |55.9         |21.2	| 78.3
> |Seqs. \#Frames > 2000|Ours     |54.2  | 63.9	     | 21.4         | 85.4
> |All Seqs.          |Trans-T |52.4  | 62.2    | 21.0         | 87.0
> |All Seqs.          |Ours     |54.9  | 65.8     | 21.4         | 85.9
>
>
>
> |Evt-LaSOT	|Method| $RSR$| $OP_{0.50}$ | $RPR_{0.075}$
> |----| ----|-----|-----|-----
> |Seqs. \#Frames < 1500|Trans-T |41.0   | 27.2    | 52.2
> |Seqs. \#Frames < 1500|Ours      |39.4   | 23.3    | 47.9
> |Seqs. \#Frames > 3000|Trans-T |27.2   |  11.0    | 18.1
> |Seqs. \#Frames > 3000|Ours 	 |31.0   | 21.5     | 34.1
> |All Seqs.                    |Trans-T |30.3  |  17.9    | 30.1
> |All Seqs.                    |Ours	 |32.3   | 22.1     | 35.9
>
> From the above tables, it can be seen that our method generally performs better on relatively short sequences than on relatively long sequences, which is also a common phenomenon for the object tracking task. On the relatively long sequences, our method consistently outperforms Trans-T to a large extent, demonstrating the advantage of our method.  Besides, in Lines 312 – 313 of the manuscript, we indeed discussed that more efforts and studies should be devoted to improving the performance on extremely long sequences, which is more challenging.
>
> **Q3**. *As far as I know, LaSOT contains 1400 sequences, and the sequence length varies from 1K to 10K frames, so how does the Evt-LaSOT construct in this article filter data from the original LaSOT? Are there specific standards? Is the filtered sequence have the same length as the original sequence?*
>
> **Response**: We randomly selected 8 sequences from the original LaSOT dataset, i.e., bird-15, bird-3, crab-6, crab-18, cat-20, cat-3, crocodile-3, bear-4. The statistics of these sequences are provided in the above table (**Response to Q2**). In the camera-ready version, we will clarify this issue.

---

> ### Author Response · Authors · 2022-08-02
> **Responses to Comments of Reviewer #kchd (Part 2)**
>
> **Q4**. *Table 2 presents the proposed algorithm has a higher FPS on Evt-LaSOT. The authors point out that the main reason is this method directly consumes raw event data without pre-processing. Is there any other reason besides this? Can you design some ablation experiments to analyze the processing speed?*
>
> **Response**: We confirm that the high efficiency of our method is mainly credited to the fact that our method directly  consumes raw event data without pre-processing. Specifically, we utilized the officially released code by FE108 for pre-processing, i.e., converting raw event data into event frames. The average time of pre-processing one event-frame is about 33.8 ms. However, for our method, such a pre-processing step is not required, and its total running time is about 39.7 ms, which validates the efficiency of our method.
>
> **Q5**. *The video demo's main screen intuitively shows this method's effect, but what do the three sub-screens on the right refer to? Please explain the specific meaning of each sub-screen.*
>
> **Response**: Sorry for the confusion caused. For the meaning of each sub-screen, we have denoted it in the first 2 seconds of the video demo, i.e., the main screen shows the tracking results (i.e., bounding boxes) of different methods, and the other three sub-screens in the right-side show, from top to the bottom, the intermediate results of motion-aware target likelihood, semantic-driven target likelihood, and motion flow.
>
> **Q6**. *Limitations: The authors discuss the public privacy issues in the additional material and argue that the event-based vision puts a lower threat to public privacy than RGB image. However, I want to know how secure and robust the event-based vision is? If we deploy the event-based scheme on self-driving systems, will it be safer than the current RGB image-based plan?*
>
> **Response**: The authors agree that RGB data-based tracking is indeed  an important part for industrial applications. However, event data-based methods have demonstrated  its potential and advantage under the special environments, e.g., strong exposure and low-light, which may further improve performance and robustness of the whole system, together with the RGB data-based tracking. The authors think that current event data-based tracking  is at the very beginning stage, compared with RGB data-based tracking, and more efforts and studies should be devoted.  Note that an Autopilot Tesla Model 3 has killed a motorcyclist on southbound I-15 at 1 a.m. Jul/24/22’, which may be caused by the poor lighting conditions in the early morning.

---

> ### Author Response · Authors · 2022-08-09
> **Thanks to kchd for Reviewing our Submission **3911**. Any More Questions?**
>
> Dear **Reviewer kchd**
>
> Thanks for your time and efforts in reviewing our submission **3911**, as well as the recognition of our work. We think we have answered your questions clearly and directly. We are also glad to answer them if you have any more questions. Thanks.
>
> The authors

---

### Official Review · Reviewer_Vqd3 · 2022-07-10

**Rating:** 5
**Confidence:** 4
**Soundness:** 3 good
**Presentation:** 2 fair
**Contribution:** 3 good

**Summary:**

Instead of converting events into 2D event frames/images, this work first sample a subset of key events and use GNN to transform them into high dimensional feature embeddings. The resulting feature embeddings are used to compute target likelihood following siamese fashion. In addition, motion-aware target likelihood is also computed to strengthen the matching process. Experiments on both synthetic and real event datasets demonstrate the its effectiveness.

**Questions:**

1. The description about key-event sampling is not very clear. For grids with empty events, do they also try to find spatially closest events as key-events?
In addition, what if there are two events that both are closest events to the grid?
Does this operation heavily compress temporal information within event streams?
2. Too many details are moved to supplementary material, which cause difficulty in understanding the whole method. For example, GNN-based spatio-temporal embedding, target likelihood prediction via motion trajectory back-tracking, and confidence-based object proposals.
3. In the experiment on synthetic data, would low fps of videos make the simulation less realistic and bring more gap between simulation and real-world data?
4. What about comparison with other event representations such as voxel grid and time surface?
5. It is not clear how to effectively supervise the motion-aware branch to get the correct motion? Is there any intermediate supervision or regularization?
6. In Fig. 3, what is the relationship between (a) (b) (c) (d)? In addition, it seems the motion vectors are noisy. Do the motion vectors can really reflect the movement triggering the corresponding key events?
7. The performance on Ev-LoSOT (PRP 30+) is very low, while the top RGB trackers (e.g. OSTrack)  have achieve performances higher than 70+ with realtime speed. It implies the experiment on Ev-LaSOT tend to fail. The SiamBAN is mentioned but is not compared in the experiments.
8. The result of FENet in Table 1 seems to be worse than the one reported in original paper.

**Limitations:**

Yes, authors have discussed the limitation of this work.

**Strengths And Weaknesses:**

1. A novel event processing method which first samples large amount of events into a subset of key events and then uses GNN to transform them into feature embeddings. the embedding representations could preserve more task-specific information than others.
2. Different from previous event-based trackers that, the proposed method explore to utilize the motion information contained within the event could, which is very interesting.

---

> ### Author Response · Authors · 2022-08-02
> **Responses to Comments of Reviewer #Vqd3 (Part 1)**
>
> Thanks for your time and efforts in reviewing our submission, as well as the valuable comments. The authors appreciate your recognition of our work. In the following, we will address all of your concerns point-by-point in detail.
>
> **Q1**. *The description about key-event sampling is not very clear. For grids with empty events, do they also try to find spatially closest events as key-events? In addition, what if there are two events that both are closest events to the grid? Does this operation heavily compress temporal information within event streams?*
>
> **Response**: As stated in Line 160 of the manuscript, for each grid point, we choose its spatially closest event as the key-event, no matter whether the grid is empty or not. For the special case that multiple events have the same distance to a typical grid point, we will randomly choose one as the key-event. Note that we use the 2D grid to sample only a few key-events efficiently and then rearrange the spatio-temporal embedding of these key-events to form a regular 2D matrix according to the indices of the grid points, which is fed into the subsequent transformer. Moreover, the learned embeddings always stay in their original positions for the subsequent key-event back-tracing and target likelihood prediction. For the potential temporal information compression, it may occur in the downsampling procedure because the spatial and temporal information is **coupled** (an event is denoted by (x, y, t)). However, it is **not  serious** for the proposed framework because the GNN-based spatio-temporal embedding utilizes the adjacent information around the key-events. More importantly, we quantitatively investigated such a compression effect via counting the ratio of perceived events (including the key events and the utilized neighbor events) by the GNN in each time interval. The following table shows that our method can perceive about 90% of events in each time interval, preserving most temporal information. Moreover, the ablation studies, i.e., 4) and 6) of Table 3 of the manuscript, also validate the effectiveness of the designed sampling algorithm.
>
> |Normalized time range    | [0,0.1) | [0.1,0.2)| [0.2,0.3)| [0.3,0.4)| [0.4,0.5)| [0.5,0.6)| [0.6,0.7)| [0.7,0.8)| [0.8,0.9)| [0.9,1.0]
> |  ---- |-----|-----|-----|-----|-----|-----|-----|-----|-----|-----
> | \#Events in event clouds	           |100301 |102215  |101460  |98821    |99518    |101035|100909	|100071 | 97715   | 97955
> | \#Events been perceived by the GNN  |88176  |92543  |93668  |92450    |93803	|95285	|94575   | 92803  | 89008     | 86761
> | Ratio of perceived events (%)	            | 87.91  | 90.54  |92.32   |93.55	|92.26	|94.31	|93.72   |92.74     |91.09       |88.57
>
> **Q2**. *Too many details are moved to supplementary material, which cause difficulty in understanding the whole method. For example, GNN-based spatio-temporal embedding, target likelihood prediction via motion trajectory back-tracking, and confidence-based object proposals.*
>
> **Response**: Sorry for the confusion caused. As claimed in the manuscript, we make **the first attempt** to construct a new end-to-end learning-based paradigm that directly consumes event clouds. Accordingly, we build the whole framework **from scratch**, rather than on the basis of an existing pipeline. Thus, there are too many technical details different from previous works. Due to space limitations, we mainly put the motivation, the function, and the brief implementations of each module in the manuscript so that the proposed paradigm can be grasped from a more global perspective. Regarding the detailed implementations, we have to move them to the supplementary material. Besides, to help the reviewers understand the method better, we also submitted the **source code**. Finally, according to the NeuIPS policy, i.e., “ If your submission is accepted, you will be allowed an additional content page for the camera-ready version,” we will add more details in the camera-ready version to make the paper better understood.

---

> ### Author Response · Authors · 2022-08-02
> **Responses to Comments of Reviewer #Vqd3 (Part 2)**
>
> **Q3**. *In the experiment on synthetic data, would low fps of videos make the simulation less realistic and bring more gap between simulation and real-world data?*
>
> **Response**: The reviewer is correct. There indeed exists a gap between the simulated and real events. Specifically, to simulate event data from the corresponding RGB video, the simulator [1] generally interpolates input videos with low FPS to ones with high FPS using estimated optical flow. Then, an event calculator is applied to the videos with high FPS to output simulated event data. As the interpolated videos lack the HDR characteristic and the employed optical flow may not be completely accurate, the simulated event data are less realistic. Despite that the simulated event data have these drawbacks, their overall characteristic and distribution are consistent with real event data, and thus such a simulation setting has been widely adopted in the field of event data processing and analysis due to the lack of real event data.
>
> **Q4**. *What about comparison with other event representations such as voxel grid and time surface?*
>
> **Response**: Note that our method is specially designed for directly processing raw event data (or event clouds), and it **cannot** consume event data with other representations, e.g., voxel grid or time surface, via post-processing. Besides, Ref. [2] has  experimentally validated that the event-frame-based representation produces better performance than time surfaces, event count, time-surface with linear time decay frames. We refer the reviewer to Table 4 J, K, L, M,and N  of Ref. [2] for more details.
>
> **Q5**. *It is not clear how to effectively supervise the motion-aware branch to get the correct motion? Is there any intermediate supervision or regularization?*
>
> **Response**: Sorry for the confusion caused. We refer the reviewer to Sec. 3 of Supplementary Material for the details of the regularization term $L_{CD}$ involved in Eq. (5) of the manuscript, which is used for supervising the flow estimation module. The event cloud within a very tiny time interval at the beginning of each event cloud sample is utilized as the potential locations of the back-traced key-events. We use Chamfer Distance to measure the discrepancy between two point sets. Moreover, considering that some key-events may be missing, or some noisy events may be sampled in such a potential target set, which may disturb the learning process, we also truncate the  accidental errors using a threshold.
>
> **Q6**. *In Fig. 3, what is the relationship between (a) (b) (c) (d)? In addition, it seems the motion vectors are noisy. Do the motion vectors can really reflect the movement triggering the corresponding key events?*
>
> **Response**: Sorry for the confusion caused. The four subfigures (a), (b), (c), and (d) indicate the tracking results of different methods on four event cloud samples from different sequences. We will update the figure caption in the camera-ready version. Besides, we also agree with the reviewer that there are indeed some noises (or errors) in the predicted motion vectors. However, the majority of the motion vectors are consistent with the movement direction of the target object (see the video demo contained in Supplementary Material). Besides, it is worth noting that most of the noisy motion vectors correspond to the noisy events. As the ground-truth motion vectors are unavailable  for this dataset, we cannot evaluate the prediction quality quantitatively.  We also want to notice that the predicted motion vectors are by-products, and although our method does not predict completely accurate motion vectors, such a motion flow estimation module **indeed makes a contribution**  (see the ablation studies in 3) and 6) of Table 3 of the manuscript). Last but not least, Fig. 3 is mainly used to demonstrate that the motion flow estimation module indeed works as what we claim, i.e., predicting motion vectors. In the future, more advanced techniques could be investigated to predict more accurate motion vectors to improve tracking performance.

---

> ### Author Response · Authors · 2022-08-02
> **Responses to Comments of Reviewer #Vqd3 (Part 3)**
>
> **Q7**. *The performance on Ev-LoSOT (PRP 30+) is very low, while the top RGB trackers (e.g. OSTrack) have achieve performances higher than 70+ with realtime speed. It implies the experiment on Ev-LaSOT tend to fail. The SiamBAN is mentioned but is not compared in the experiments.*
>
> **Response**: As mentioned in the **Response to Q3**, the simulated event data may be less realistic (lack of sufficient information, compared with real event data). As stated in Lines 277-280 of the manuscript, considering that this issue may compromise the performance of event tracking methods, resulting in unfair comparisons with RGB-trackers, we only compared with event tracking methods  in order to demonstrate the “relative superiority” of the proposed method  used for tracking **non-rigid** objects in the experiments conducted on the simulated event dataset Evt-LaSOT (Note that all target objects in the real event dataset FE108 are rigid). Besides, the SOTA RGB-trackers utilize larger-scale training datasets, e.g., TrackingNet, LaSOT, COCO, and GOT-10K, which is also unfair to event-based methods.  For the method SiamBAN, according to the results shown in Table 2 of Ref. [2], it can be known that SiamBAN does not perform well for event data. The comparison with the more powerful Transformer-tracker [3] is more convincing to demonstrate the advantage of the proposed method. Actually, when preparing for the manuscript, considering the space limitation we removed the SiamBAN but forgot to delete it in the final manuscript. Sorry for the mistake. We will revise it in the camera-ready version.
>
> **Q8**. *The result of FENet in Table 1 seems to be worse than the one reported in the original paper.*
>
> **Response**: Note that FENet is a multi-modality method, which consumes both RGB and event data to achieve tracking. As mentioned in **Line 249** of our manuscript, for a fair comparison, we compared the proposed method only with the event branch of FENet (its performance is listed in Table 4 B of Ref. [2]). Besides, we also want to note that we retrained the event-branch of FENet and obtained a higher RSR (52.3) than that reported in the original paper (i.e., 52.0). Thus, we confirm the results of FENet are correct.
>
> [1] Gehrig D, Gehrig M, Hidalgo-Carrió J, et al. Video to events: Recycling video datasets for event cameras, in Proc.  IEEE/CVF CVPR, 2020, pp. 3586-3595.
>
> [2] Zhang J., Yang X., Fu Y., et al. Object tracking by jointly exploiting frame and event domain, in Proc. IEEE/CVF ICCV, 2021, pp. 13043-13052.
>
> [3] Chen X, Yan B, Zhu J, et al. Transformer tracking, in Proc. IEEE/CVF CVPR, 2021, pp. 8126-8135.

---

> ### Author Response · Authors · 2022-08-09
> **Thanks to Vqd3 for Reviewing our Submission **3911**. Any More Questions?**
>
> Dear **Reviewer Vqd3**
>
> Thanks for your time and efforts in reviewing our submission **3911**, as well as the recognition of our work. We think we have answered your questions clearly and directly. We are also glad to answer them if you have any more questions. Thanks.
>
> The authors

---

### Official Review · Reviewer_bkyP · 2022-07-11

**Rating:** 5
**Confidence:** 4
**Soundness:** 3 good
**Presentation:** 3 good
**Contribution:** 3 good

**Summary:**

This paper make the first attempt to construct a new end-to-end learning-based paradigm that directly consumes event clouds. Considered the high temporal resolution and sparse characteristics in event clouds, it proposed key-event and graph-based network to extract irregular spatio-temporal information from raw data. Then this paper proposes a dual-path architecture to predict the possibility of a key-event of the search event cloud. The dual-path architecture captures the similarity of each search key-event to the template key-events and utilize the information of motion trajectories respectively. This paper demonstrates the superiority of the proposed framework over state-of-the-art methods in terms of both the tracking accuracy and speed.

**Questions:**

(1) Motivation mentioned the high temporal resolution and sparse characteristics of event data. Where is the specific design for these two characteristics the data processing and model design.
(2) The ablation experiment did not discuss how much does the use of raw data contributed to the final results. Does removing the uniquely designed data processing part of your method make a big difference to the final result ?
(3) When comparing with SOTA, why do you choose some RGB-based methods to train yourself instead of comparing with other event-based methods?
(4) “Considering that trajectories are usually continuous, the former event points should have relatively smaller moving offsets” . The offset is derived from the projection of the key-event in the last step, and the key-events in the last step is not the same timestamp either.  And the time difference between the same key-event before and after is not sure. Is it unreasonable to have a relationship between the displacement and the timestamp?


**Ethics Review Area:**

["I don’t know"]

**Limitations:**

The motivation is very convincing, but targeted design and description with respect to motivation are needed. The comparison experiment should also be focused on the points emphasized in the motivation.

**Strengths And Weaknesses:**

(1) The paper has a clear structure and is easy to understand the motivation and innovation.
(2) It proposed key-event and graph-based network to extract irregular spatio-temporal information from raw data, Making better use of the characteristics of event data.
(3) It developing a new end-to-end learning-based paradigm that directly consumes raw event data , in which new modules considering the unique features of event clouds.

Weaknesses：
(1) The specific technical details are not clear enough
(2) This paper mentioned the high temporal resolution and sparse characteristics of event data. The biggest motivation of this paper is to design data feature extraction module and algorithm model according to the unique attributes of data. But I didn't see a targeted design of the unique attributes of the Event data.

---

> ### Author Response · Authors · 2022-08-02
> **Responses to Comments of Reviewer #bkyP (Part 1)**
>
> Thanks for your time and efforts in reviewing our submission, as well as the valuable comments. The authors appreciate your recognition of our work. In the following, we will address all of your concerns point-by-point in detail.
>
> **Q1**. *Motivation mentioned the high temporal resolution and sparse characteristics of event data. Where is the specific design for these two characteristics the data processing and model design.*
>
> **Response**: The traditional event frame-based trackers accumulate events into event-frames, inevitably compromising the temporal resolution, while the proposed method directly consumes raw event data (or event clouds), which can naturally preserve original high temporal resolution. Moreover, the sparse and asynchronous sensing manner (i.e., sparsity) results in that the underlying structure of event clouds is irregular (Lines 134 – 135 of the manuscript).  To deal with such an irregular characteristic, we propose to adopt the graph neural network (GNN) to embed the spatio-temporal information. Compared with the event frame-based method, which convolves on both triggered and inactivated pixels, our GNN-based method directly processes the sparse event points, which is more plausible for event data processing. Besides, to handle the huge data samples caused by the raw event data efficiently, we propose a simple yet effective downsampling strategy. Finally, the subsequent modules, including target likelihood prediction via semantic-driven Siamese-matching and motion trajectory back-tracking, object proposal, as well as the training strategies, are all specially designed to adapt to event cloud data. Note that in Table 3 of the manuscript, we experimentally validated the effectiveness of each module.
>
> **Q2**. *The ablation experiment did not discuss how much does the use of raw data contributed to the final results. Does removing the uniquely designed data processing part of your method make a big difference to the final result?*
>
> **Response**: First it is worth noting that all modules and strategies of our framework are specifically and carefully designed for **directly processing raw** event data (or event cloud data), e.g., the key-event sampling algorithm, graph neural network-based key-event embedding, key-event back-tracing, etc. Thus, the proposed method **cannot** be fed with other types of event representations. But we believe the advantage of using raw event data could  be validated to some extent, based on the following facts: the methods under comparison, including FENet [1], E-PrDiMP and E-TransT that handle **event-frame**, are built upon RGB data-based pipelines, which contain many **advanced and mature** techniques, while our framework is built **from scratch** and thus relatively plain.  In this situation, our method still achieves better performance than the compared methods (see Table 1 of the manuscript), demonstrating the great potential of directly processing raw event data, which retains the unique characteristics of event data to the greatest extent. Besides, we conducted comprehensive ablative studies in Table 3 of the manuscript, which convincingly validate the effectiveness of each of the  specially designed modules.

---

> ### Author Response · Authors · 2022-08-02
> **Responses to Comments of Reviewer #bkyP (Part 2)**
>
> **Q3**. *When comparing with SOTA, why do you choose some RGB-based methods to train yourself instead of comparing with other event-based methods?*
>
> **Response**: To the best of our knowledge, when submitting our manuscript, FENet [1] compared in Table 3 of the manuscript is the only officially published end-to-end learning-based event tracking work, whose source code is also publicly available.
>
> - We did not compare with model (or non-learning) -based methods because most of them  did not release their source codes. Besides, for model-based methods, hyperparameters have to be tuned for each specific sequence, making it hard to achieve desired performance on the large-scale dataset.
>
> - RGB data-based tracking has been studied for many years with great efforts devoted, and many mature techniques have been proposed, producing impressive performance. Thus, we believe it is convincing to construct event tracking baselines by modifying the strong RGB-trackers to adapt event data. Note that such a manner was also adopted in Ref. [1]. As shown in Table 3 of the manuscript, the modified event-trackers from RGB-trackers, i.e., E-PrDiMP and E-TransT, which consume event-frames,  achieve reasonable performance.
>
> - Finally, we also want to notice that the leftmost four methods in Table 3, i.e., ATOM, DiMP, SiamFC++, and PrDiMP, take the corresponding RGB data as input, which are provided to illustrate the differences between RGB data-based tracking and event data-based tracking.
>
> **Q4**. *The offset is derived from the projection of the key-event in the last step, and the key-events in the last step is not the same timestamp either. And the time difference between the same key-event before and after is not sure. Is it unreasonable to have a relationship between the displacement and the timestamp?*
>
> **Response**: Sorry for the confusion caused. The key-event back-tracing module is designed to back-trace each key-event to the place, where it should be, in the 2-D initial x-y plane whose time is the initial time of the corresponding event cloud sample. Thus, we can compare back-traced key-events with the previous bounding box, as those key-events have been coordinated to be with the same timestamp as the bounding-box. Besides, the time difference could be easily calculated, i.e., the current key-event time-stamp minus the initial time of this event cloud sample.
>
> [1] Zhang J, Yang X, Fu Y, et al. Object tracking by jointly exploiting frame and event domain, in Proc. IEEE/CVF  ICCV, 2021, pp. 13043-13052.

---

> ### Author Response · Authors · 2022-08-09
> **Thanks to Reviewer bkyP for Reviewing our Submission **3911**.  Any More Questions?**
>
> Dear **Reviewer bkyP**,
>
> Thanks for your time and efforts in reviewing our submission **3911**, as well as the recognition of our work. We think we have answered your questions clearly and directly. We are also glad to answer them if you have any more questions. Thanks.
>
> The authors

---

### Official Review · Reviewer_mj7a · 2022-07-11

**Rating:** 6
**Confidence:** 4
**Soundness:** 3 good
**Presentation:** 2 fair
**Contribution:** 3 good

**Summary:**

The paper tackles the problem of event data-based object tracking. Instead of using event frame/image representation as in the existing methods, the authors chose to process raw event clouds for event data-based object tracking. The proposed framework includes a downsampling strategy to extract key events from a raw event cloud, a graph-based network to embed spatio-temporal information of key-events, semantic and Motion-aware Target Likelihood Prediction, and a confidence-based object proposal. The authors conduct experiments on FE108 [49] and a new synthetic dataset Event LaSOT for tracking the evaluation of non-rigid objects. All the baselines and the proposed method are trained on FE108 and evaluated on the testing set of FE108 and Evt-LaSOT. The experiments prove the possibility of consuming raw event clouds for object tracking and achieving state-of-the-art performance on challenging datasets.

**Questions:**

The current recommendation is Borderline accept (5). Overall, the authors indeed prove the claim that they design an architecture that consumes raw event cloud and achieves promising object tracking performance on two challenging datasets compared with STOA. The paper could be improved by discussing the difference between the proposed methods with existing 3D point cloud tracking literature. Moreover, it would be interesting to demonstrate that existing 3D point cloud trackers could not outperform the proposed method as it considers the characteristics of the event cloud.

**Limitations:**

Yes, the authors discuss the limitations of the work.

**Strengths And Weaknesses:**

Strengths:
1. The authors focus on event-based object tracking, and we make the first attempt to construct a new end-to-end learning-based paradigm that directly consumes event clouds. The reviewer found the direction will interest those working on event-based cameras. Moreover, those who work on point cloud modeling might find a new problem to test the applicability and generalization of their methods.
2. The work contributes a network architecture that can consume a raw event cloud and demonstrate that the network can outperform other event cloud representations for object tracking on two challenging datasets.
3. Convincing tracking results are demonstrated in the supplementary material.
4. The implementation of this framework has been released.

Weaknesses:
1. 3D point cloud tracking: The proposed framework shares similarities with existing 3D point cloud tracking literature, e.g., Qi et al., 2020 and Zhou et al., 2022. It will be essential to summarize the difference between the literature pools. Moreover, the authors could analyze how the proposed architecture harvests characteristics of event cloud (e.g., inherent motion trajectories) where methods built for 3D point cloud tracking do not consider these aspects. Moreover, the experimental section could be further improved by comparing the proposed framework with a state-of-the-art 3D point cloud tracker, which aims to show the importance of considering data characteristics.
- Qi et al., P2B: Point-to-Box Network for 3D Object Tracking in Point Clouds, CVPR 2020
- Zhou et al., PTTR: Relational 3D Point Cloud Object Tracking with Transformer, CVPR 2022

2. Ablative studies: The reviewer found the contribution of the two target likelihood predictors unclear. Specifically, based on the implementation of the proposed architecture, the authors could have removed one of the predictions and tested the corresponding model. However, in Table 3, the performance of removing semantic-aware target likelihood prediction is missing.

3. STOA comparison: In Table 1, the proposed method demonstrates inferior performance compared to E-PrDiMP on the HDR setting. The discussion in Sec. 4.2 does not explicitly discuss it. It would be essential for the authors to provide the corresponding insights as they mention that event-based generally surpass RGB-based methods under LL and HDR scenarios.

---

> ### Author Response · Authors · 2022-08-02
> **Responses to Comments of Reviewer #mj7a (Part 1)**
>
> Thanks for your time and efforts in reviewing our submission, as well as the valuable comments. The authors appreciate your recognition of our work. In the following, we will address all of your concerns point-by-point in detail.
>
> **Q1**. *3D point cloud tracking: The proposed framework shares similarities with existing 3D point cloud tracking literature …. comparing the proposed framework with a state-of-the-art 3D point cloud tracker, which aims to show the importance of considering data characteristics.*
>
> **Response**: We agree with the reviewer that the raw event data (or event clouds) could be thought of as 3D point cloud (PC) data. However, they are still different due to the fact that these two kinds of data have different physical meanings. Specifically, 3D Point clouds by the LIDAR sensor record the geometric information of 3d entities, which are generally collected by measuring how long the emitted pulses of infrared light take to come back after hitting nearby objects/scenes. Only the part of the object/scene visible to the LIDAR sensor will be perceived (one emitted pulse will generate at most one 3D point). Thus, the captured 3D points are distributed on the **surface** of the 3D entity.    However, even data are acquired by measuring the pixel intensity variations in a short time period, and for a typical pixel, there may be multiple events (i.e., the events have the same x and y dimensions but different t dimensions). Thus, the events are distributed **within a 3D cube**. Such a data distribution difference may make it **inappropriate** to apply PC-trackers to event data. Considering that the final object proposal of the event data is in the 2D spatial domain, which is different from that of 3D point cloud data, we design a key-event sampling algorithm to enlarge the spatial receptive field, as mentioned in **Lines 137 – 139** of our manuscript, which is also experimentally validated in  4) and 6) of Table 3 of the manuscript.
>
> Besides, the high-temporal resolution of event data naturally retains the object motion information. The proposed event-tracker is able to utilize such inherent motion information to back-trace the potential initial positions of key-events and learns a motion-aware target likelihood for boosting the classic semantic-driven Siamese-matching, which is also different from current PC-trackers. Moreover, to avoid accumulated errors by such a motion-aware target likelihood estimation module, we also design a robust training-strategy by injecting noise into the fed bounding-box, as mentioned in **Lines 137 – 139** of our manuscript (experimentally validated in the 3) and 2) in Table 3 of the manuscript).
>
> Following your suggestion, we also conducted experiments to compare the proposed method with SOTA PC-trackers on the FE108 dataset. Specifically, we trained the PC-trackers with the same experiment settings as ours, and we also tried our best to adapt those methods to event data for optimal performance. As shown in the following table, it can be seen that the proposed method exceeds the PC-trackers [1, 2] to a large extent, i.e., 7.4% and 15.5% improvements in terms of the success rate and precision respectively, which demonstrates the demand of designing event-specialized tracking algorithms. We will add some discussions about the relation between event- and PC-trackers in the camera-ready version.
>
> |Methods|  $RSR$ | $OP_{0.50}$| $OP_{0.75}$| $RPR$
> |----| ----| ----- |-----|-----
> |P2B CVPR’20 [1]	 |36.1	| 35.4 	       | 4.7        	  |40.5
> |PTTR CVPR’22 [2]	 |47.5	| 49.6          | 11.5      	  |70.2
> |Ours     		 |54.9   | 65.8          | 21.4         |85.9
>
> **Q2**. *Ablative studies: The reviewer found the contribution of the two target likelihood predictors unclear. Specifically, based on the implementation of the proposed architecture, the authors could have removed one of the predictions and tested the corresponding model. However, in Table 3, the performance of removing semantic-aware target likelihood prediction is missing.*
>
> **Response**:  In the ablative study (Table 3 of the manuscript), we always reserve the semantic-aware target likelihood module under different cases because we think it  is a basic and necessary module in the framework to achieve object tracking. Following your suggestion, we also conducted such an ablation study on the FE108 dataset, i.e., removing this module with all the other modules unchanged.   As listed in the following table, it can be seen that the tracking performance decreases significantly.
>
> |  |  $RSR$ | $OP_{0.50}$| $OP_{0.75}$| $RPR$
> |----|----|-----|-----|-----
> |W/O semantic-aware target likelihood|45.2   | 43.5	       | 10.2         | 65.3
> |Full model                 			 |54.9   | 65.8          | 21.4         | 85.9

---

> > ### Comment · Reviewer_mj7a · 2022-08-08
> > **Responses to the authors' feedback to the three questions (all resolved)**
> >
> > Thanks for the authors’ time and efforts in responding to my questions.
> >
> > 1. As written in the Section of “Question,” the reviewer’s main suggestion is to explore if the existing 3D point cloud trackers can be applied for this task:
> >     - The authors describe the difference between the two data types (3D point cloud acquired by laser scanners vs. event cloud acquired by event cameras).
> >     - The authors implement the two recent point cloud trackers (i.e., [1] and [2]) on the event camera data.
> >     - The experiments show that the proposed tracker outperforms existing point cloud trackers by a large margin.
> > Empirically, the numbers show the importance of considering data characteristics. While the results are promising, the reviewer would encourage the authors to incorporate further discussions on what we could do to sort out all factors (e.g., network architectures) that could potentially contribute to the performance gain such that we can identify the "actual" gain given by the proposed sampling strategy.
> >
> > 2. What is the importance of the proposed *semantic-aware target likelihood prediction? Thanks for the additional ablative study. The experiment indeed shows that semantic-aware target likelihood prediction plays an essential role.*
> > 3. Thanks for the clarification. My concern is resolved.

---

> > > ### Author Response · Authors · 2022-08-09
> > > **Thanks to Reviewer mj7a for acknowledging our responses and the additional suggestion**
> > >
> > > The authors are glad to know that all your concerns have been resolved. Besides, thanks for your additional suggestion. Here, we also provide more explanations, which will be discussed in the camera-ready version.
> > >
> > > **Q1.** *Sorting out all factors (e.g., network architectures) that could potentially contribute to the performance gain to identify the "actual" gain from the sampling strategy.*
> > >
> > > **Response**: Currently, we demonstrate the effectiveness and advantage of the proposed sampling strategy based on the  ablative study results shown in 4) and 6) of Table 3 of the manuscript, where in 4), random sampling is used to replace the proposed sampling strategy with the remaining settings exactly the same as 6), i.e., the full model.  Qualitatively, the proposed sampling method may sample more regularly distributed key-events than the random sampling, which indeed may facilitate the following swin-layer embedding.
> > >
> > > More importantly, following your suggestion, we also designed a **plain model** to **exclude** the potential effects of other architectures such that the effectiveness of the sampling layer can be uniquely identified. Specifically, **the plain model only contains the basic and necessary modules**, i.e., 4 hieratical graph convolution layers to embed event clouds, a self-attention layer to globally fuse embeddings, a cross-attention layer to match the embeddings from template to search key-events (the semantic-driven Siamese-matching branch), and a target proposal module same as the proposed method. We trained and tested the plain model with the same experiment settings but different sampling methods. As listed in the following table, it can be seen that the plain model with the proposed sampling strategy achieves 3.1% and 4.4% improvements in terms of RSR and RPR, respectively, compared with the plain model with random sampling. Since both the graph convolution and attention layers are widely adopted for point set processing, we think that such a plain model **has no bias** to different sampling methods, and thus, the effectiveness and advantage of the proposed sampling strategy can be uniquely validated. We will add more discussions in the camera-ready version.
> > >
> > > |Methods|$RSR$ | $OP_{0.50}$|$OP_{0.75}$|$RPR$
> > > |----|----|-----|-----|-----
> > > |The plain model with  random sampling |44.1|43.7|9.7|65.1
> > > |The plain model with  proposed sampling|47.2|49.1|11.0|69.5

---

> > > > ### Comment · Reviewer_mj7a · 2022-08-09
> > > > **Thank you for the responses**
> > > >
> > > > Thank you for your time and effort in this response. The additional analysis is helpful, and detailed explanations are convincing. Please incorporate the discussions in the final version if the paper is accepted. Thank you so much for the extraordinary efforts and for sharing your insights on handling event clouds for object tracking. I have updated my rating accordingly.

---

> > > > > ### Author Response · Authors · 2022-08-09
> > > > > **Thanks to Reviewer mj7a for the valuable comments.**
> > > > >
> > > > > The authors greatly appreciate Reviewer mj7a for the thoughtful and valuable comments. The authors also believe that event data has strong potential for object tracking! Thanks very much!

---

> ### Author Response · Authors · 2022-08-02
> **Responses to Comments of Reviewer #mj7a (Part 2)**
>
> **Q3**. *STOA comparison: In Table 1, the proposed method demonstrates inferior performance compared to E-PrDiMP on the HDR setting …. event-based generally surpass RGB-based methods under LL and HDR scenarios.*
>
> **Response**: Note that E-PrDiMP refers to the modified PrDiMP (an RGB-based tracker) to adapt to event data in the form of the frame representation. In the HDR (high dynamic range) scenario of the FE108 dataset, the objects have relatively slow movement, making that the edges of the objects in the event-frame are relatively clear. Thus, the event frame-based method E-PrDiMP may grasp the semantic meaning of objects better, leading to its higher tracking performance. Meanwhile, the higher tracking performance of  E-PrDiMP than PrDiMP in the LL (low-light) and HDR scenarios also supports our statement that “event-based generally surpass RGB-based methods under LL and HDR scenarios”. Such an advantage is credited to the high dynamic range characteristic of the event camera, as mentioned in Lines 30 – 32 of our manuscript, i.e., it could capture pixel intensity variation under very poor light conditions, while traditional RGB cameras may even have no response under these conditions. We also refer the reviewer to the Fig.7 of Ref. [3] for the visual comparison between event and RGB data. We will add the corresponding discussion in the camera-ready version.
>
> [1] Qi H., Feng C., Cao Z., et al. P2b: Point-to-box network for 3d object tracking in point clouds, in Proc. IEEE/CVF CVPR, 2020, pp. 6329-6338.
>
> [2] Zhou C., Luo Z., Luo Y., et al. PTTR: Relational 3D Point Cloud Object Tracking with Transformer, in Proc.  IEEE/CVF CVPR, 2022, pp. 8531-8540.
>
> [3] Zhang J., Yang X., Fu Y., et al. Object tracking by jointly exploiting frame and event domain, in Proc. IEEE/CVF ICCV, 2021, pp. 13043-13052.

---

### Author Response · Authors · 2022-08-05
**Any More Questions? Let's Discuss Them. Thanks.**

Dear Reviewers

Thanks for your time and efforts in reviewing our submission **3911** and the valuable comments. We hope our detailed responses have addressed all your concerns. We are glad to answer them if there are any more questions.

We are looking forward to your responses.


The authors

---

### Meta-Review · Area_Chair_N2a8 · 2022-08-27

**Recommendation:** Accept
**Confidence:** Certain

**Metareview:**

The paper receives overall positive reviews and rebuttal has resolved the reviewer's concerns. The paper proposes a new framework that directly takes raw event clouds as inputs for object tracking. Reviewers agree that this innovation is inspiring. AC agrees and recommends accepting the paper.

**Award:**

No

---

### Decision · Program_Chairs · 2022-09-14

Accept